# Exogenous Application of Amino Acids Mitigates the Deleterious Effects of Salt Stress on Soybean Plants

Kolima Peña Calzada [1,2,*,†], Dilier Olivera Viciedo [3,*,†], Eduardo Habermann [4],
Alexander Calero Hurtado [1,2], Priscila Lupino Gratão [5], Renato De Mello Prado [1], Luis Felipe Lata-Tenesaca [6],
Carlos Alberto Martinez [4], Gabriela Eugenia Ajila Celi [5] and Juan Carlos Rodríguez [2]

1 Laboratory of Plant Nutrition, Soils and Fertilizers Sector, Department of Agricultural Production Sciences, São Paulo State University "Júlio de Mesquita Filho" (UNESP), Jaboticabal 14884-900, São Paulo, Brazil
2 Agronomy Department, University of Sancti Spiritus "Jose Marti Perez" (UNISS), Sancti Spiritus 60100, Cuba
3 Center of Environment and Agriculture Science, Federal University of Maranhão, Rodovia BR 222, km 4, s/n, Chapadinha 65500-000, Maranhão, Brazil
4 Department of Biology, FFCLRP, University of Sao Paulo (USP), Ribeirão Preto 14040-900, São Paulo, Brazil
5 Department of Biology, Faculty of Agrarian and Veterinary Sciences, São Paulo State University (UNESP), Jaboticabal, Via de Acesso Prof. Paulo Donato Castelane, S/N, Vila Industrial, Jaboticabal 14884-900, São Paulo, Brazil
6 Department of Plant Pathology, Federal University of Viçosa, Viçosa 14884-900, Minas Gerais, Brazil
* Correspondence: kolimapena@gmail.com (K.P.C.); dilierolvi@gmail.com (D.O.V.)
† These authors contributed equally to this work.

**Abstract:** The cultivated area of soybean has increased worldwide in past decades, including regions with saline soils, strongly decreasing growth and productivity. The use of amino acids (AAs) as buffering compounds against stressful conditions can be a useful strategy to mitigate salt stress in these regions. This study aimed to evaluate the effects of foliar application of AA mixtures on the growth, physiology, and biochemistry traits of salt-stressed soybean plants. A pot experiment was designed as a factorial scheme ($4 \times 3$) in a randomized complete design (RCD). Treatments consisted of four concentrations of AA mixtures of a non-VA application, 0.4 mL L$^{-1}$, 0.8 mL L$^{-1}$, and 1.2 mL L$^{-1}$ (VIUSID Agro® (VA) source), which were combined with non-salt stressed and salt-stressed groups (50 and 100 mmol L$^{-1}$ NaCl), to analyze improvement in growth and potassium (K$^+$) accumulation, maintenance of relative water content (RWC), net photosynthesis rate ($A$), transpiration ($E$), stomatal conductance ($gs$), and chlorophyll content, and increase of proline accumulation and water use efficiency (iWUE). Moderate and high salinity induced a notable increase in oxidative and ionic biomarkers, coupled with higher Malondialdehyde (MDA) concentration and Na$^+$ accumulation. Alternatively, soybean growth, K$^+$ accumulation, and physiological and biochemical parameters were decreased under salinity. Foliar spraying of AAs drastically increased osmolyte accumulation associated with sustained iWUE and RWC, increased proline accumulation, and improved $A$, $E$, $gs$, and chlorophyll content. Greater outcomes were achieved with the foliar spraying of amino acids at 1.2 mL L$^{-1}$. Collectively, foliar application of AA mixtures plays an important role in salt stress remediation by modifying important physiological and biochemical processes, thereby resulting in a higher growth of soybean plants.

**Keywords:** attenuation compounds; foliar application; abiotic stress; *Glycine max*; salinity; Viusid agro

## 1. Introduction

Agricultural sustainability is being threatened by the potential negative impacts of climate change on crops [1] as well as plant homeostatic instability as result of global warming and water and nutrient restrictions [2,3]. Inadequate strategies of irrigation are increasing soil salinity levels [4,5]; salt stress is one of the main abiotic factors that impairs soybean productivity around the world [6]. Salinity soils account up to 20% of agricultural

lands in the world, with an expectation that this will increase to 50% by the end of 21st century [7].

According to the literature, crops growing in salinity soils are subject to osmotic stress, poor physical soil conditions, nutritional disturbances, toxicity, and reduced productivity. Limiting crop losses due to salt stress is a major concern within the context of growing food demand [8]. Plant responses to salt stress depend on the salt concentration and the type of salt [9]. One of the first responses to salt stress is a decrease in the rate of leaf surface expansion, followed by stomatal closure and a decrease of photosynthesis and *E* rates [10]. In addition, a reduction in chlorophyll content, chloroplast functioning, and plant growth are observed [4,9].

Plant responses to salt stress are diverse and impact many different biochemical and physiological processes. In general, salt stress reduces leaf area, photosynthesis, and *E* due to stomatal closure [5]. In addition, there is an increased production of reactive oxygen species (ROS) due to the disruption of cellular homeostasis, leading to damage to biomolecules such as lipids, DNA, and proteins [11]. Reactive oxygen species (ROS) scavenging occurs through the plant antioxidant defense system, which is composed of enzymatic and non-enzymatic compounds [9,12]. The tolerance of salt stress is widely dependent on species and stress intensity; soybean is considered a moderately tolerant species with regard to salinity stress [13].

Plants develop many different mechanisms to cope with salt stress conditions. Among these tolerance mechanisms are the upregulation of the antioxidant defense system, efficient ion exclusion, and accumulation of osmolytes and secondary metabolites [14]. As an important response to stress factors, plants possess specific adaptive physio-molecular responses, i.e., osmotic adjustment (OA) and enhanced antioxidant capacity [15]. OA results from the assimilation of several osmolytes, such as proline and glycine betaine and inorganic ions [16]. Another crucial plant tolerance strategy under salt stress is the activation of the defense system against oxidative damage. The antioxidant system includes superoxide dismutase, catalase, peroxidases, reductases, ascorbic acid, glutathione, polyphenols, etc. [9,17]. Salinity stress interrupts ion homeostasis, resulting in a buildup of noxious ions in plants [14]. Additionally, increasing $K^+$ uptake helps to maintain ion homeostasis and to regulate the osmotic balance, to maintain turgor and to regulate the membrane potential, cytoplasmic homeostasis, protein synthesis, and enzyme activation [9]. However, $K^+$ content in plant tissues progressively decreases with increasing salinity [18]. Therefore, maintaining an adequate level of $K^+$ is essential for the survival of plants under salt stress. Plant ability to decrease the $Na^+$ uptake, maintaining an adequate concentration of $K^+$ in the cytoplasm, can contribute to plant tolerance and improved growth under salt stress conditions [18,19].

The exogenous application of AAs is reported to have positive effects on plant growth and development under stressful conditions [20,21]. Exogenous application of arginine and glycine in maize plants under stress improved plant growth [20]. In addition, arginine restored leaf relative water content and increased proline (Pro) content, improving water status and reducing oxidative stress in wheat seedlings under stress conditions [22]. Moreover, the concentration of photosynthetic pigments, shoot length, stem diameter, number of leaves, and shoot fresh and dry mass increased significantly in sunflower plants under salt stress treated with arginine [23]. In lettuce plants submitted to salt stress, tryptophan increased leaf number, leaf and root dry biomass, and total plant leaf area [24].

Although the use of AAs as buffering compounds against salt stress is a promising technique, there are only a few reports regarding the use of AA mixtures (aspartic acid, arginine, glycine, and tryptophan) in attenuating the effects of salt stress in soybean plants [25]. On the other hand, soybean is one of the main oilseed crops, with a global planted area of 127.842 million hectares and a production of 362.947 million tons in the 2020/2021 harvest. It is of great importance for Brazilian agriculture, being the main agribusiness crop in the country. In fact, Brazil stands out as one of the largest producers and is responsible for the production of 135.409 million tons over an area of 38,502 million

hectares, resulting in an average productivity of 3.517 kg/ha. Soybean cultivation is the main activity responsible for the expansion of the agricultural frontier in the country, mainly in the Cerrado region in the Brazilian northeast, which has the highest concentration of salinity soils and low rainfall [26].

Therefore, this study aimed to evaluate the effects of foliar spraying of AA mixtures on the growth, physiology, and biochemistry traits of salt-stressed soybean plants. Here, we tested two hypotheses: (i) foliar application of AA mixtures can regulate salinity-induced changes for adequate physiological and biochemical adaptations in soybean plants and (ii) higher concentrations of AA mixtures are more promising in attenuating salinity effects in salt-stressed soybean plants.

## 2. Materials and Methods

A pot experiment in a glass greenhouse was conducted in the Department of Agricultural Biology at São Paulo State University (UNESP), Jaboticabal, Brazil. Seeds of soybean (*Glycine max* (L.) Merr) cv. M-Soy 8222 were sown in pots of 5 dm$^3$ containing washed sand of medium texture as a substrate. Sand was previously washed in the following sequence: running water, deionized water, hydrochloric acid 1% solution, and deionized water [27]. The experiment was performed with two plants per pot. After seedling emergence, a Hoagland and Arnold nutrient solution (NS) was applied [28]. A concentration of nutritive solution of 10% was used six days after the emergence of seedlings, and a concentration of 75% until the end of the experimental period. The pH of the NS was adjusted to $5.7 \pm 0.2$ using HCl (1.0 mol L$^{-1}$).

### 2.1. Climate Data

The climate conditions monitored during the experiment are shown in Figure 1. Air temperature and relative air humidity in the moment of amino acid application were 23.9 °C and 86.2%, respectively.

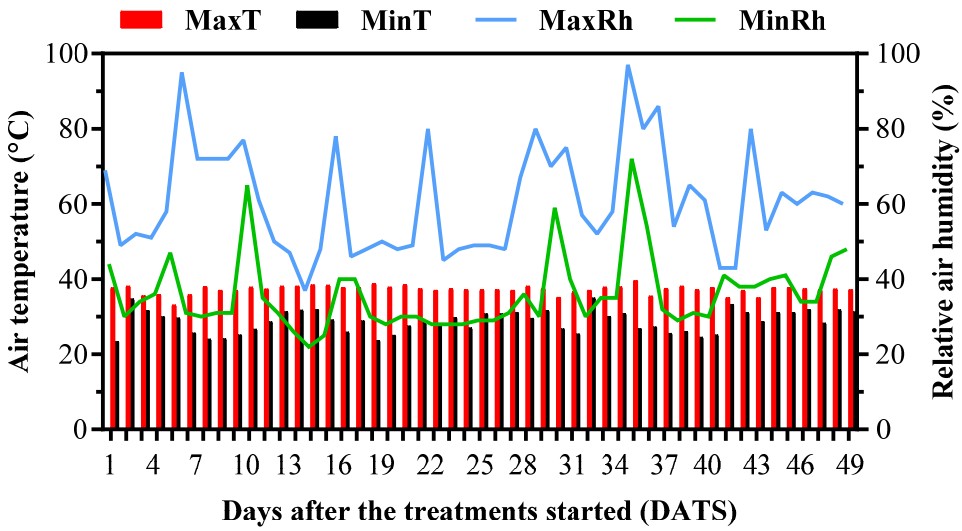

**Figure 1.** Climate data registered during the experimental period. Maximum air temperature (MaxT) (°C), minimum air temperature (MinT) (°C), maximum relative air humidity (MaxRh) (%), minimum relative air humidity (MinRh) (%).

### 2.2. Experimental Design

During the 49 days of the experimental period, pot experiments were arranged factorially (4 × 3) in a complete randomized design (CRD), with five replicates (n = 5), giving a total of 60 pots with two soybean plants. Treatments consisted of four concentrations of AA mixtures, with a non-VA application (VA0), 0.4 mL L$^{-1}$ (VA1), 0.8 mL L$^{-1}$ (VA2), and 1.2 mL L$^{-1}$ (VA3) (VIUSID Agro® (VA) source), which were combined with non-salt stressed and salt-stressed groups (50 and 100 mmol L$^{-1}$ NaCl). VIUSID Agro® product was

used as AA treatments and was applied via foliar applications. It contained AA mixtures such as glycine (2.5%), arginine (2.4%), aspartic acid (1.6%), and tryptophane (0.5%) (further details provided in Supplementary Table S1).

### 2.3. Amino Acid and NaCl Treatments

Five days after emergence, soybean plants were acclimatized with Hoagland nutrient solution for 15 days. After 15 days of adaptation, salt treatment was initiated and maintained in the corresponding pots by adding NaCl (EC 5 and 10 dS m$^{-1}$) every day through Hoagland nutrient solution until the end of the experiment. Foliar spraying of VA was diluted in deionized water and applied four times in the fully developed leaves at phenological stages V3, V4, and V5, as they are the ones with maximum growth in soybean plants [29], with an interval of 7 days. During foliar application of VA, the substrate was covered with paper towel to prevent the sprayed VA solution from coming in contact with the substrate. The NaCl solution was applied daily together with Hoagland nutrient solution. The NaCl stress and foliar spraying of AAs began in stage V3 and ended in V5.

### 2.4. Sampling Date

After 25 days from the final AA mixture spraying (at full bloom, 50 days from emerging), soybean plants were harvested for morpho-physiological, biochemical, and molecular assessment. After plant harvest, plant material was washed with detergent solution 0.2%, hydrochloric acid 0.1%, and deionized water twice to eliminate all residues [30].

#### 2.4.1. Growth Parameters

After 50 days of the experimental period, morphological parameters were evaluated in terms of leaf dry mass (g per plant, LDM), stem dry mass (g per plant, SDM), root dry mass (g per plant, RDM), and leaf area (cm$^2$, LA). Plant material was separated into leaves (without the petioles), stems, and roots and placed inside paper bags. Bags were maintained inside an oven at 65 °C until a constant dry weight was achieved for LDM, SDM, and RDM determination. LA was measured by using a leaf area meter (Ll-3100; LICOR, Inc., Lincoln, NE, USA).

#### 2.4.2. Leaf Gas Exchange

Leaf gas exchange was measure 13 days after the treatments started using an infrared gas analyzer LCPro-SD (ADC BioScientific, Hoddesdon, UK). Measurements were performed between 9:30 am and 11:30 am in one expanded leaf per plant (2 plants per plot), located in the third node from the top to bottom with ambient $CO_2$ concentration, radiation of 2.000 mol m$^{-2}$ s$^{-1}$, and leaf temperature of 26 °C. Net photosynthesis rate (*A*), stomatal conductance (*gs*), and leaf transpiration rate (*E*) were measured. Water use efficiency (iWUE) was calculated as the ratio of *A* to *E* (*A/E*).

#### 2.4.3. Chlorophyll Fluorescence

Maximum quantum efficiency of photosystem II (Fv/Fm) and photochemistry of PSII (Fv/F0) were measured at 14 days after the treatment started in two first expanded leaves from bottom to top per plant at 8:00 am using a portable fluorometer Opti Sciences Os30P. Leaves were dark-acclimated for 30 min before the measurements [31].

#### 2.4.4. Leaf Temperature

Leaf temperature was measured in three middle expanded leaves per pot 15 days after the treatments started using an infrared thermometer Testo 835-H1. Measurements were performed at midday (between 12:00 p.m. and 13:00 p.m.) on a day with no clouds, avoiding possible interference of shading in leaf temperature measurements. We used an emissivity (ε) value of 0.98.

### 2.4.5. Leaf Chlorophyll Index

We measured the leaf chlorophyll index (CCM chlorophyll index) using a CCM-200 Chlorophyll Meter (Opti-Science, Hudson, NH, USA) at the predawn period (5:00 a.m.–6:00 a.m.) and midday period (between 12:00 p.m. and 13:00 p.m.). Measurements were conducted on the adaxial surface of 3 expanded leaves per pot 15 days after the treatments started.

### 2.4.6. Relative Water Content (RWC)

The RWC of the plant leaves was determined using the methods as described Barrs and Weatherley [32]. Briefly, 10 leaf discs (1 cm in diameter) were taken from the fully expanded leaves. The fresh weight (FW) was recorded, and then the discs were incubated in distilled water for 4 h. Turgid weight (TW) was recorded, and the discs were put in an oven at 70 °C for 24 h to calculate the dry weight (DW). RWC was measured as follows:

$$RWC\ (\%) = (FW - DW)/\ (TW - DW) \times 100$$

### 2.4.7. Determination of $Na^+$ and $K^+$ Accumulation

The concentration of $Na^+$ and $K^+$ was measured according to Bataglia et al. [33]. Finely pulverized leaf samples were digested with a mixture of nitric acid ($HNO_3$) and perchloric acid ($HClO_4$) in a 3:1 ratio for $Na^+$ and $K^+$ concentration analysis. The concentrations of $Na^+$ and $K^+$ in the digested mixture were estimated by flame photometry (Jencon PFP 7; JENCONS-PLS, Bed- fordshire, England). Based on the concentration of $Na^+$ and $K^+$ in leaves of soybean plants and their respective leaf dry matter, the accumulation of this nutrient was calculated and expressed as follows (mg $g^{-1}$).

### 2.4.8. Malondialdehyde (MDA) Concentration

Lipid peroxidation was assessed by measuring the malondialdehyde (MDA) concentration ($\mu$mol mg$^{-1}$ FW) using thiobarbituric acid-reactive substances (TBARS), following the previously described procedure by Mihara et al. [34]. Two hundred milligrams of leaf and root tissue was homogenized in 20% trichloroacetic acid (TCA) and centrifuged at $14,000 \times g$ for 20 min at 4 °C. Equal amounts of supernatant and 0.5% thiobarbituric acid (TBA) in 20% TCA ($w/v$) and 0.01% butylated hydroxytoluene (BHT) were mixed. The mixture was incubated in boiling water at 95 °C for 20 min, after which the reaction was stopped by placing the tubes in an ice bath. The samples were centrifuged at $14,000 \times g$ for 15 min to produce a clear solution. The absorbance of the supernatant was read at 532 nm using a spectrophotometer (Beckman DU 640, San Diego, CA, USA). The reading at 600 nm was subtracted to remove nonspecific absorption. The concentration of LPO was calculated from the extinction coefficient (155 mM$^{-1}$ cm$^{-1}$).

### 2.4.9. Proline Concentration

Leaves and roots were collected for proline concentration (mg $g^{-1}$ FW) analysis. Samples were immediately placed in liquid nitrogen and stored at $-80$ °C until analysis. Leaf proline concentration was measured according to the acid ninhydrin method [35].

### 2.5. Statistical Analysis

The data presented in this manuscript were from one typical experiment, and statistical analysis was conducted using factorial analysis of variance (ANOVA) ($p < 0.05$) to test the effects of the four concentrations of s and three levels of salinity stress and their interactions (VA $\times$ NaCl). Datasets were checked for normality and homogeneity by Shapiro-Wilk and Levene tests, respectively, to meet ANOVA assumptions. Average values were compared by a least significant difference (LSD) test ($p < 0.05$). The Statistical Package AgroEstat$^{\circledR}$ was used to perform these tests [28]. Additionality, the method of orthogonal polynomials was used to identify functional relationships (linear or quadratic) between response NaCl and VA treatments on LT, using GraphPad Prism 8.03 software (GraphPad Software, Inc., San Diego, CA, USA). Data were presented as means $\pm$ standard error (SE) of five replications.

## 3. Results

### 3.1. Growth Character Measurement

We observed significant interaction ($p < 0.01$) between NaCl and VA on LDM, SDM, RDM, and LA (Figure 2a–d). Salinity stress considerably decreased dry mass (leaf, stem, and roots) and LA by 17%, 14%, 16%, and 19% under 50 mmol $L^{-1}$ and 13%, 18%, 18%, and 10% under 100 mmol $L^{-1}$, respectively, relative to the non-NaCl treated plants ($p < 0.001$); however, this inhibition was mitigated by VA application, particularly at higher concentrations (VA3, 1.2 mL $L^{-1}$) (Figure 2a–d). In non-NaCl stressed soybean plants, the plant growth was higher in all VA concentrations applied, especially at higher concentrations (1.2 mL $L^{-1}$ of VA) compared to the VA0Na0 treatment ($p < 0.01$) (Figure 2a–d). Different VA concentration applications evidenced the helpful impacts on plant growth. With foliar spraying of VA3, LDM, SDM, and RDM, as well as LA, were significantly higher than that in salt-affected plants ($p < 0.01$), specifically by 18%, 29%, 17%, and 27% under 50 mmol $L^{-1}$, and by 15%, 11%, 19%, and 13% under 100 mmol $L^{-1}$, respectively. These results indicated that AA mixture application plays an important role in enhancing plant growth under salinity conditions.

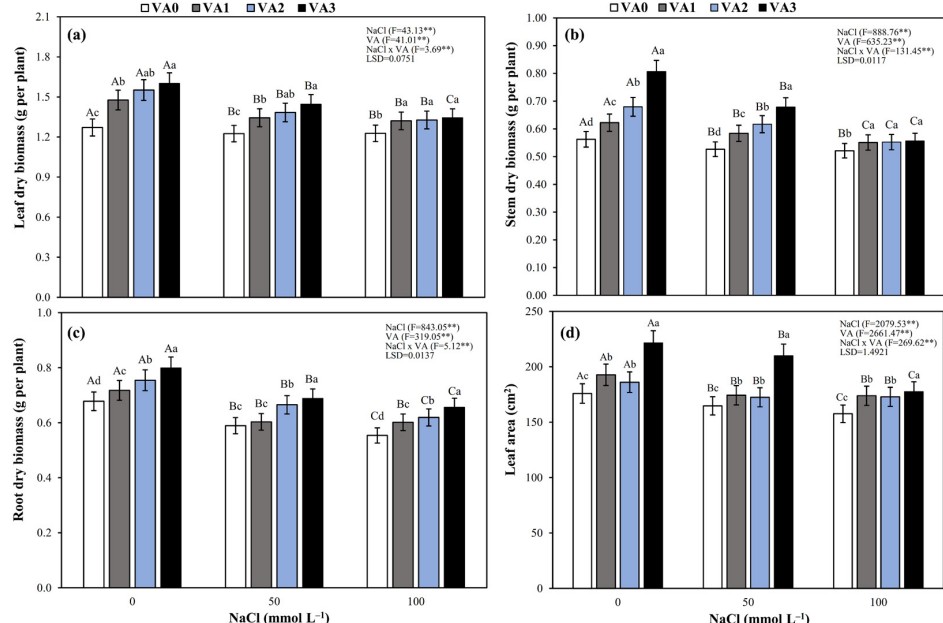

**Figure 2.** Leaf (**a**), stem (**b**), and root (**c**) dry mass and leaf area (**d**) of soybean plants under different levels of salt stress and VA application. Plants were submitted to four concentrations of VA (VA0, non-VA application; VA1, 0.4 mL $L^{-1}$; VA2, 0.8 mL $L^{-1}$; and VA3, 1.2 mL $L^{-1}$) under three levels of NaCl (0, 50, and 100 mmol $L^{-1}$). Data are mean $\pm$ SD (n = 5). Uppercase letters above bars indicate significant differences between different NaCl treatments at the same VA concentration. Lowercase letters above bars indicate differences among VA concentrations at the same NaCl concentration, according to LSD test ($p < 0.05$). F values from ANOVA: ** $p < 0.01$.

### 3.2. Leaf Gas Exchange

A means comparison revealed significant ($p < 0.01$) interaction between VA and NaCl stress was significant on all leaf gas exchange parameters ($A$, $E$, and $g_s$) measured in soybean plants (Figure 3a–c). Under moderate (50 mmol $L^{-1}$ NaCl) and high (100 mmol $L^{-1}$ NaCl) salt stress, the $A$, $E$, and $g_s$ were dramatically decreased ($p < 0.011$). Salt stress greatly decreased $A$ in all NaCl concentrations, especially at the highest levels (100 mmol $L^{-1}$ NaCl). However, VA application greatly mitigated the negative impacts of salt stress on soybean plants. In non-NaCl stress conditions, the $A$, $E$, and $g_s$ were higher in all VA treatments, especially at higher concentration (VA3, 1.2 mL $L^{-1}$), relative to the non-VA application ($p < 0.01$) (Figure 3a–c). Under moderate NaCl stress (50 mmol $L^{-1}$), the higher

*A* values were achieved under VA2 and VA3 concentrations applied to plants compared to the non-VA application and VA1 treatment; however, this last treatment showed significant difference ($p < 0.01$) and higher effects on *A* in comparison to the VA0 treatment. Nevertheless, under higher NaCl stress (100 mmol L$^{-1}$), all VA treatments promoted higher *A* compared to the non-VA application ($p < 0.01$) (Figure 3a). Salinity stress significantly ($p < 0.01$) decreased *E* as NaCl concentration increased, regardless of VA concentration application (Figure 3b). Conversely, *E* was improved by all VA treatments under salt stress (50 and 100 mmol L$^{-1}$ NaCl), particularly under V2 and V3 treatments, and showed significant difference ($p < 0.02$) compared to VA0 treatment and VA1 treatment; however, this last treatment showed higher values in comparison with the non-VA application (Figure 3b). Salt stress (50 and 100 mmol L$^{-1}$ NaCl) treatments caused a significant ($p < 0.01$) decrease in $g_s$ of soybean plants (Figure 3c). Compared to the VA0 and VA1 treatments, higher $g_s$ was achieved in all VA concentrations applied, especially under VA2 and VA3 treatments, and the most effective concentrations were under 50 mmol L$^{-1}$ of NaCl; however, VA1 treatment showed higher $g_s$ than that non-VA application (Figure 3c). Under high salinity (100 mmol L$^{-1}$ of NaCl), the most favorable for increasing $g_s$ was the VA3 treatment ($p < 0.01$); however, VA1 and VA2 concentration showed similar effects on $g_s$ and higher responses compared to the VA0 treatment (Figure 3c). This supports the idea that changes in leaf gas exchange under salt stress conditions are triggered by the foliar application of AA mixtures.

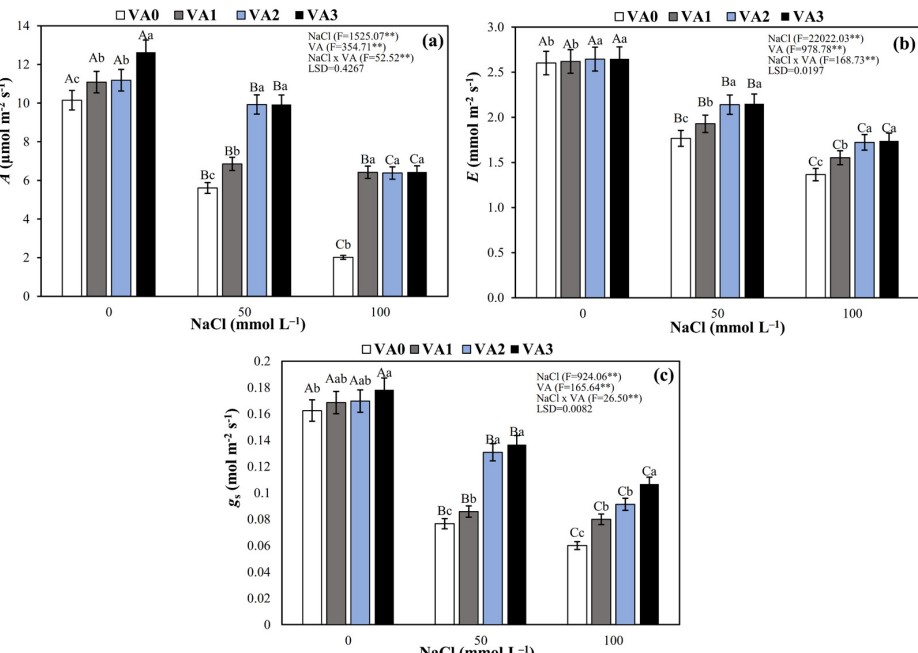

**Figure 3.** Leaf gas exchange of soybean leaves under different levels of salt stress and VA application. Net photosynthesis rate (*A*) (**a**), leaf transpiration rate (*E*) (**b**) and stomatal conductance ($g_s$) (**c**). Plants were submitted to four concentrations of VA (VA0, non-VA application; VA1, 0.4 mL L$^{-1}$; VA2, 0.8 mL L$^{-1}$; and VA3, 1.2 mL L$^{-1}$) under three levels of NaCl (0, 50, and 100 mmol L$^{-1}$). Data are mean ± SD (n = 5). Uppercase letters above bars indicate significant differences between different NaCl treatments at the same VA concentration. Lowercase letters above bars indicate differences among VA concentrations at the same NaCl concentration, according to LSD test ($p < 0.05$). F values from ANOVA: ** $p < 0.01$.

### 3.3. Leaf Temperature (LT) and Leaf Chlorophyll Index (LCi)

The ANOVA showed a significant interaction between NaCl and VA treatments on LT and LCi (Figure 4a,b) and leaf chlorophyll index (Figure 4b). Using a correlation study involving NaCl and VA treatments, we were able to evaluate the strategies of LT of soybean plants. LT increased with increasing NaCl stress (Figure 4a). Conversely, VA application

showed a significantly decreased quadratic response in LT under non-NaCl stress, which decreased linearly with increasing VA concentration under 50 and 100 mmol L$^{-1}$ of NaCl, particularly in the VA3 treatment (Figure 4a). A significant decline in LCi was observed with increasing salinity stress (Figure 4b). Conversely, a pronounced acceleration in LCi was observed in soybean plants grown at different VA treatments under non-salt stress and salt stress conditions. Maximum LCi concentrations were observed in soybean plants at VA2 (0.8 mL L$^{-1}$) and VA3 (1.2-mL L$^{-1}$) treatments under non-salt stress and 50 mmol L$^{-1}$, and showed significant difference ($p < 0.01$) compared to the other VA treatments (Figure 4b). All VA treatments greatly increased in LCi under 100 mmol L$^{-1}$, as compared to the VA0 treatments (Figure 4b). These changes in LT and LCi by foliar application of AA mixtures could be an important mechanism to increase salt tolerance in plant species.

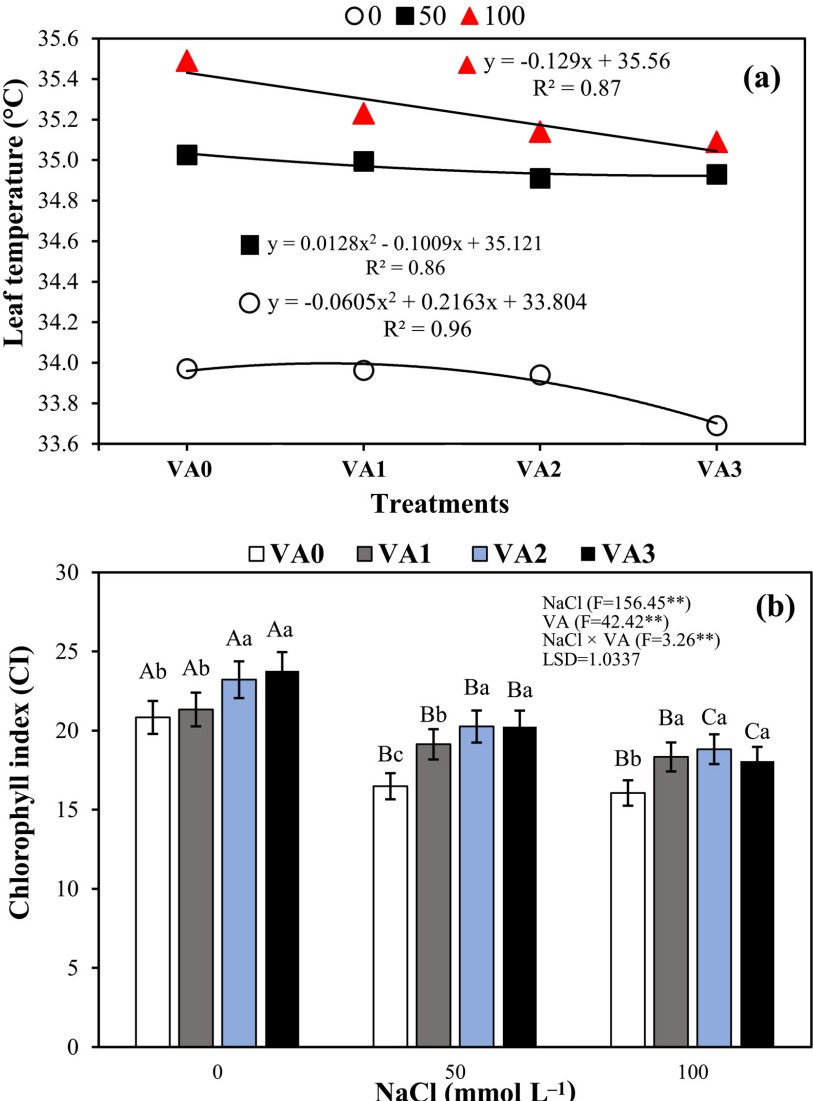

**Figure 4.** Leaf temperature (**a**) and leaf chlorophyll index (**b**) of soybean plants under different levels of salt stress and VA application. Plants were submitted to four concentrations of VA (VA0, non-VA application; VA1, 0.4 mL L$^{-1}$; VA2, 0.8 mL L$^{-1}$; and VA3, 1.2 mL L$^{-1}$) and three levels of NaCl (0, 50, and 100 mmol L$^{-1}$). Data are mean $\pm$ SD (n = 5). In Figure 4b, uppercase letters above bars indicate significant differences among NaCl concentrations at the same VA concentration. Lowercase letters above bars indicate differences among VA concentrations at the same NaCl concentration, according to LSD test ($p < 0.05$). F values from ANOVA: ** $p < 0.01$. Concentrations of 100, 50, and 0 mmol L$^{-1}$ of NaCl are represented by (▲), (■), (○), respectively.

### 3.4. Maximum Quantum Efficiency of PSII (Fv/Fm) and Photochemistry Efficiency of PSII (Fv/Fo)

There was a significant ($p < 0.01$) and interactive effect of NaCl and VA on the photochemical efficiency of PSII (Fv/Fm) and photochemistry of PSII (Fv/F0) in soybean plants Figure 5a,b). Salinity stress (50 and 100 mmol $L^{-1}$ NaCl) decreased Fv/Fm and Fv/F0, but these effects were reversed by all VA concentrations applied, especially for the VA3 concentration under 50 mmol $L^{-1}$ NaCl, and VA2 and VA3 treatments in improving Fv/Fm and Fv/F0 and showed significant difference ($p < 0.03$) compared to the VA0 and VA1 treatments. However, at the same time, this last treatment showed higher Fv/Fm and Fv/F0 relative to the non-VA application (Figure 5a,b). These results indicate the potential of foliar spraying of AA mixtures to counteract the harmful effects of salinity.

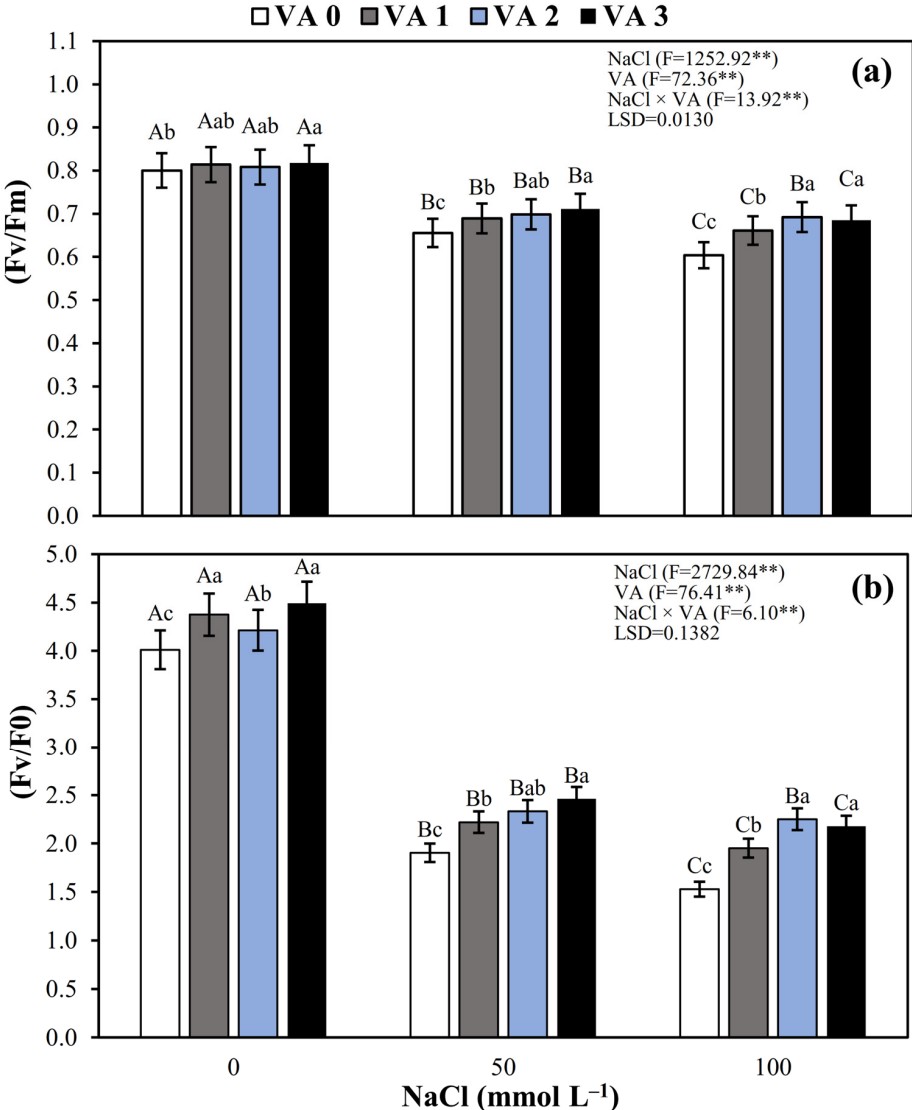

**Figure 5.** Maximum quantum efficiency of PSII (Fv/Fm) (**a**) and photochemistry efficiency of PSII (Fv/Fo) (**b**) of soybean plants under different levels of salt stress and VA application. Plants were submitted to four concentrations of VA (VA0, non-VA application; VA1, 0.4 mL $L^{-1}$; VA2, 0.8 mL $L^{-1}$; and VA3, 1.2 mL $L^{-1}$) and three levels of NaCl (0, 50, and 100 mmol $L^{-1}$). Data are mean $\pm$ SD (n = 5). Uppercase letters above bars indicate significant differences among NaCl concentrations at the same VA concentration. Lowercase letters above bars indicate differences among VA concentrations at the same NaCl concentration, according to LSD test ($p < 0.05$). F values from ANOVA: ** $p < 0.01$.

### 3.5. Relative Water Content (RWC) and Water-Use Efficiency ($_i$WUE)

ANOVA revealed significant ($p < 0.01$) interaction between NaCl and VA treatments on leaf *RWC* and *iWUE* of soybean plants (Figure 6a,b). Under non-salt stress conditions, VA1, VA2, and VA3 treatments increased RWC by 7, 12, and 10%, respectively, as compared to VA0 treatment ($p < 0.001$) (Figure 6a). Leaf water status was affected by salt stress, as confirmed by the lower RWC values observed in treatments with 50 and 100 mmol L$^{-1}$ of NaCl. In soybean plants, under non-salt stress conditions, all VA applications increased the leaf RWC compared to the VA0 treatment ($p < 0.002$), particularly with the VA2 and VA3 concentrations. Under moderate salt stress (50 mmol L$^{-1}$ of NaCl), the leaf RWC was higher in the VA2 and VA3 treatments and was significantly higher ($p < 0.001$) than VA0 and VA1 treatments. Additionally, under high salt stress (100 mmol L$^{-1}$ of NaCl), the VA3 treatment resulted in higher leaf RWC compared to the other VA treatments; nevertheless, the RWC in the leaves of soybean plants was similar and higher in the VA1 and VA2 treatments compared to the VA0 treatment ($p < 0.001$) (Figure 6a).

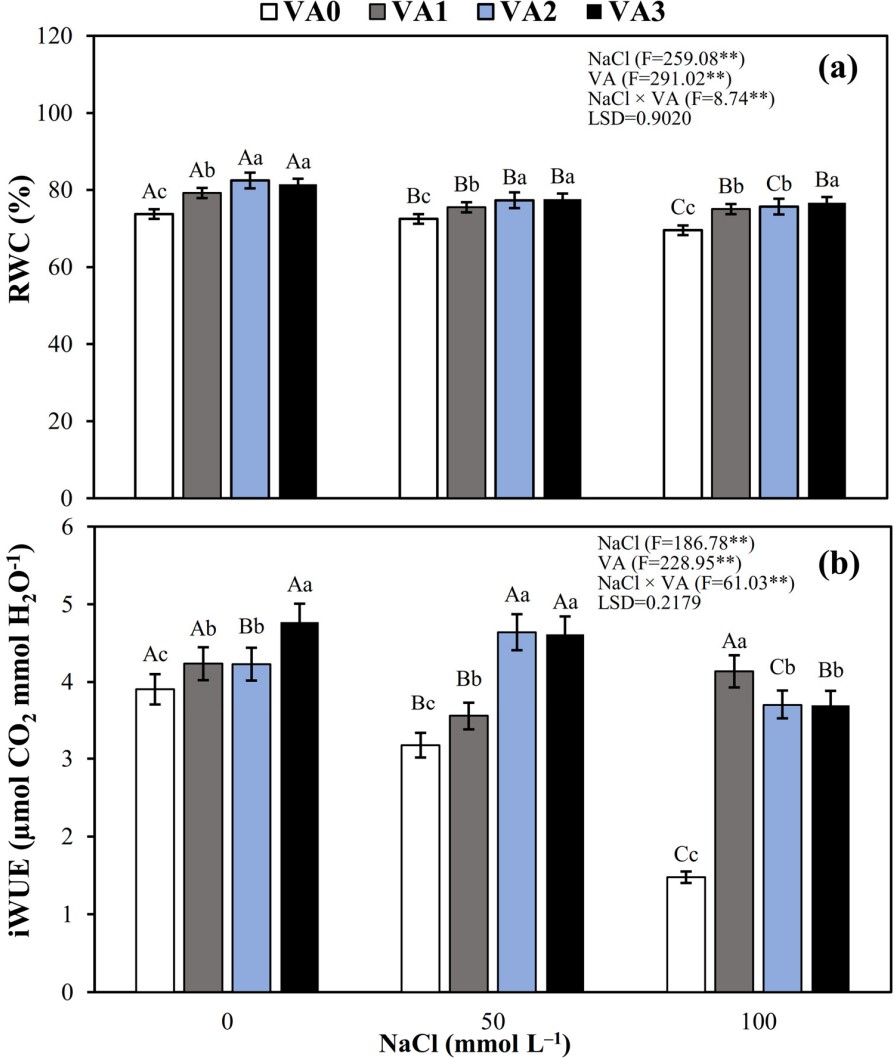

**Figure 6.** Relative water content (RWC) (**a**) and water-use efficiency (iWUE) (**b**) of soybean plants under different levels of salt stress and VA applications. Plants were submitted to four concentrations of VA (VA0, non-VA application; VA1, 0.4 mL L$^{-1}$; VA2, 0.8 mL L$^{-1}$; and VA3, 1.2 mL L$^{-1}$) under three levels of NaCl (0, 50, and 100 mmol L$^{-1}$). Data are mean ± SD (n = 5). Uppercase letters above bars indicate significant differences among NaCl concentrations at the same VA concentration. Lowercase letters above bars indicate differences among VA concentrations at the same NaCl concentration, according to LSD test ($p < 0.05$). F values from ANOVA: ** $p < 0.01$.

The experimental results showed that the iWUE of soybean leaves was significantly ($p < 0.001$) affected by salt stress salinity (Figure 6b). VA application also caused a marked increase in iWUE, both under salinity and non-salinity conditions; however, in this last condition, the higher iWUE was achieve in the VA3 treatment as compared to the other VA applications. Additionality, both VA1 and VA2 treatments increased iWUE and showed significant difference ($p < 0.02$) compared to the VA0 treatment (Figure 6b). Nevertheless, the higher iWUE under 100 mmol $L^{-1}$ of NaCl was achieved in the VA2 and VA3 treatments in comparison with the other VA treatments, but foliar application of VA1 showed higher iWUE than that in the VA0 treatment (Figure 6b). On the other hand, under 100 mmol $L^{-1}$ of NaCl, we found higher iWUE in the VA1 treatment compared to the other VA treatments; both VA2 and VA3 showed similar effects and higher iWUE values than the VA0 application ($p < 0.002$) (Figure 6b).

### 3.6. Na$^+$ and K$^+$ Accumulation

ANOVA revealed interaction effects between NaCl and VA on Na$^+$ and K$^+$ accumulation in leaf and roots of soybean plants ($p < 0.003$) (Figure 7a–d). Salt stress conditions increased Na$^+$ accumulation and decreased K$^+$ accumulation, but these effects were reversed by the foliar spraying of VA. In the absence of salt stress (0 mmol $L^{-1}$ of NaCl), all VA concentrations had no effect on the Na$^+$ accumulation in leaves and roots (Figure 7a,b). However, under 50 mmol $L^{-1}$ of NaCl, all VA concentrations applied decreased Na$^+$ accumulation, especially with the VA3 treatment, which showed lower Na$^+$ accumulation in leaves and roots by 29% and 19%, respectively, as compared to the VA0 treatments. Additionally, VA1 and VA2 showed equal effects ($p < 0.738$) and lowered Na$^+$ accumulation by 20% and 18% in leaves and roots, respectively ($p < 0.001$) (Figure 7a,b).

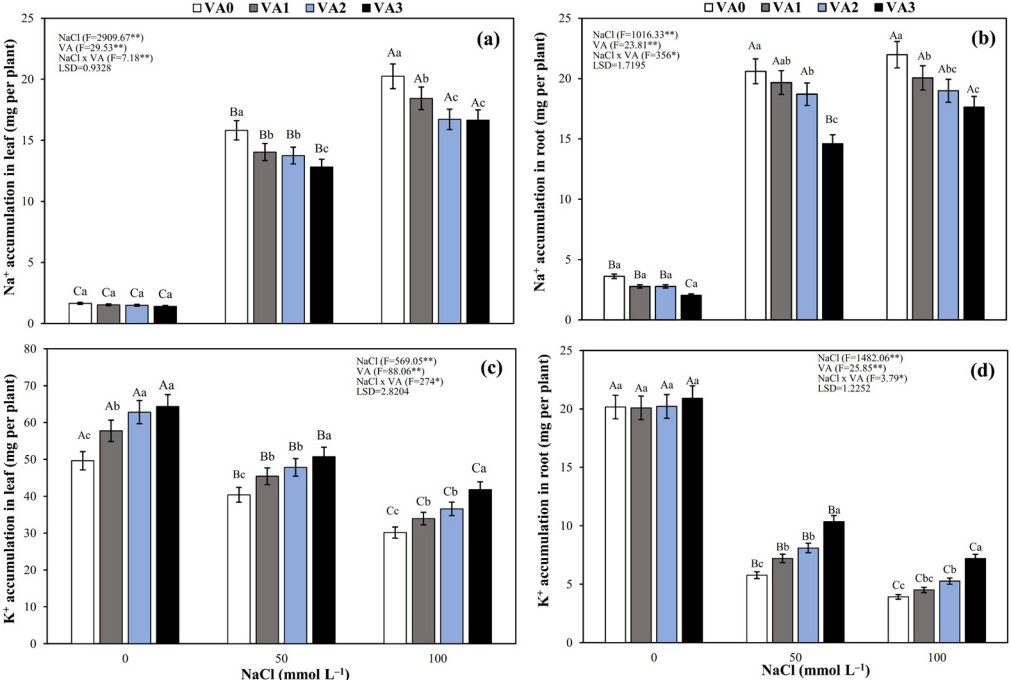

**Figure 7.** Na$^+$ content in leaves (**a**) and roots (**b**); K$^+$ content in leaves (**c**) and roots (**d**) of soybean plants under different levels of salt stress and VA applications. Plants were submitted to four concentrations of VA (VA0, non-VA application; VA1, 0.4 mL $L^{-1}$; VA2, 0.8 mL $L^{-1}$; and VA3, 1.2 mL $L^{-1}$) and three levels of NaCl (0, 50, and 100 mmol $L^{-1}$). Data are mean $\pm$ SD (n = 5). Uppercase letters above bars indicate significant differences among NaCl concentrations at the same VA concentration. Lowercase letters above bars indicate differences among VA concentrations at the same NaCl concentration, according to LSD test ($p < 0.05$). F values from ANOVA: ** $p < 0.01$, * $p < 0.05$.

Salt stress dramatically decreased the $K^+$ accumulation in leaves by 71% and 81% and in roots by 17% and 25% under 50 and 100 mmol $L^{-1}$ of NaCl, respectively, and showed significant difference ($p < 0.003$) compared to the non-NaCl treatment. While VA application drastically increased levels in leaves and roots of soybean plants (Figure 7c,d). However, $K^+$ accumulation under 50 mmol $L^{-1}$ of NaCl was higher in the VA3 treatment by 79% in leaves and 26% in roots; whereas in 100 mmol $L^{-1}$ of NaCl, the increases were 84% and 39% in leaves and roots, respectively, compared to the VA0 treatments ($p < 0.001$). In addition, VA1 and VA2 treatments showed similar effects and increased $K^+$ accumulation by 33% in leaf and 18% in roots under 50 mmol $L^{-1}$ of NaCl and by 34% and 17% in leaves and roots, respectively, under high salinity conditions (100 mmol $L^{-1}$ of NaCl), and showed a significant difference ($p < 0.001$) compared to the VA0 treatment (Figure 7c,d). Thus, our results suggest that a key mechanism for the attenuation of salt stress in soybean plants by foliar spraying of AA mixtures is the inhibitory effect of $Na^+$ uptake and increasing $K^+$ accumulation.

### 3.7. Leaf and Root MDA Concentration

We observed interaction effects ($p < 0.002$) between NaCl and VA on leaf and root MDA concentration (Figure 8a,b). In both organs, the concentration of MDA increased with increasing NaCl concentration in the nutrient solution (Figure 8a,b). Conversely, MDA concentration was significantly lower in all VA treatments, especially in leaves of soybean at VA3 concentration under moderate salt stress (50 mmol $L^{-1}$), which showed significant difference ($p < 0.01$) compared to the others VA treatments assessed; however, the lower concentration of MDA under high salt stress (100 mmol $L^{-1}$) conditions was achieved in VA1 and VA2 treatments in comparison with the VA0 and VA3 treatments. Nevertheless, this last treatment showed lower MDA concentration than that the VA0 treatment ($p < 0.001$) (Figure 8a). In addition, root MDA concentration under 50 mmol $L^{-1}$ of NaCl conditions was similar and lower in VA2 and VA3 treatments compared to the other VA treatments studied; however, root MDA concentration under 100 mmol $L^{-1}$ of NaCl was lower in the VA3 concentration in comparison with the other VA treatments, and there was a lower concentration of MDA in the roots of soybean plants in VA2 and VA1 treatments than in the VA0 treatment ($p < 0.001$) (Figure 8b). These results evidenced that foliar application of AA mixtures can regulate the response of lipid peroxidation in different levels of salinity stress.

### 3.8. Leaf and Root Proline Concentration

There is a significant ($p < 0.01$) interaction between NaCl and VA on leaf proline concentration (LPC) and root proline concentration (RPC) (Figure 9a,b). LPC and RPC increased in soybean plants with increasing NaCl stress (Figure 9a,b). In addition, foliar spraying of VA further increased LPC and RPC; however, LPC under non-salt stress (0 mmol $L^{-1}$ of NaCl) VA treatments had no effect ($p = 0.0882$) (Figure 9a). In addition, LPC was higher in VA3 treatments by 65% under 50 mmol $L^{-1}$ of NaCl and 24% under 100 mmol $L^{-1}$ of NaCl and showed significant difference ($p < 0.001$) compared to the other VA concentrations, but the VA2 concentration under 50 mmol $L^{-1}$ of NaCl and VA1 and VA2 treatments under 100 mmol $L^{-1}$ of NaCl increased LPC more than the VA1 and VA0 treatments, respectively (Figure 9a). Similar increases in RPC were achieved in VA2 and VA3 treatments (with significant difference; $p < 0.01$) by 24%, as compared with the VA0 and VA1 treatments under the three NaCl levels studied; however, the application of the VA1 treatment significantly increased RPC by 19% in comparison with the VA0 treatments under moderate and high salt stress conditions $p < 0.01$) (Figure 9b). These results indicated that foliar application of AA mixtures plays an important role in finding an alternative approach for enhancing salt tolerance in plants.

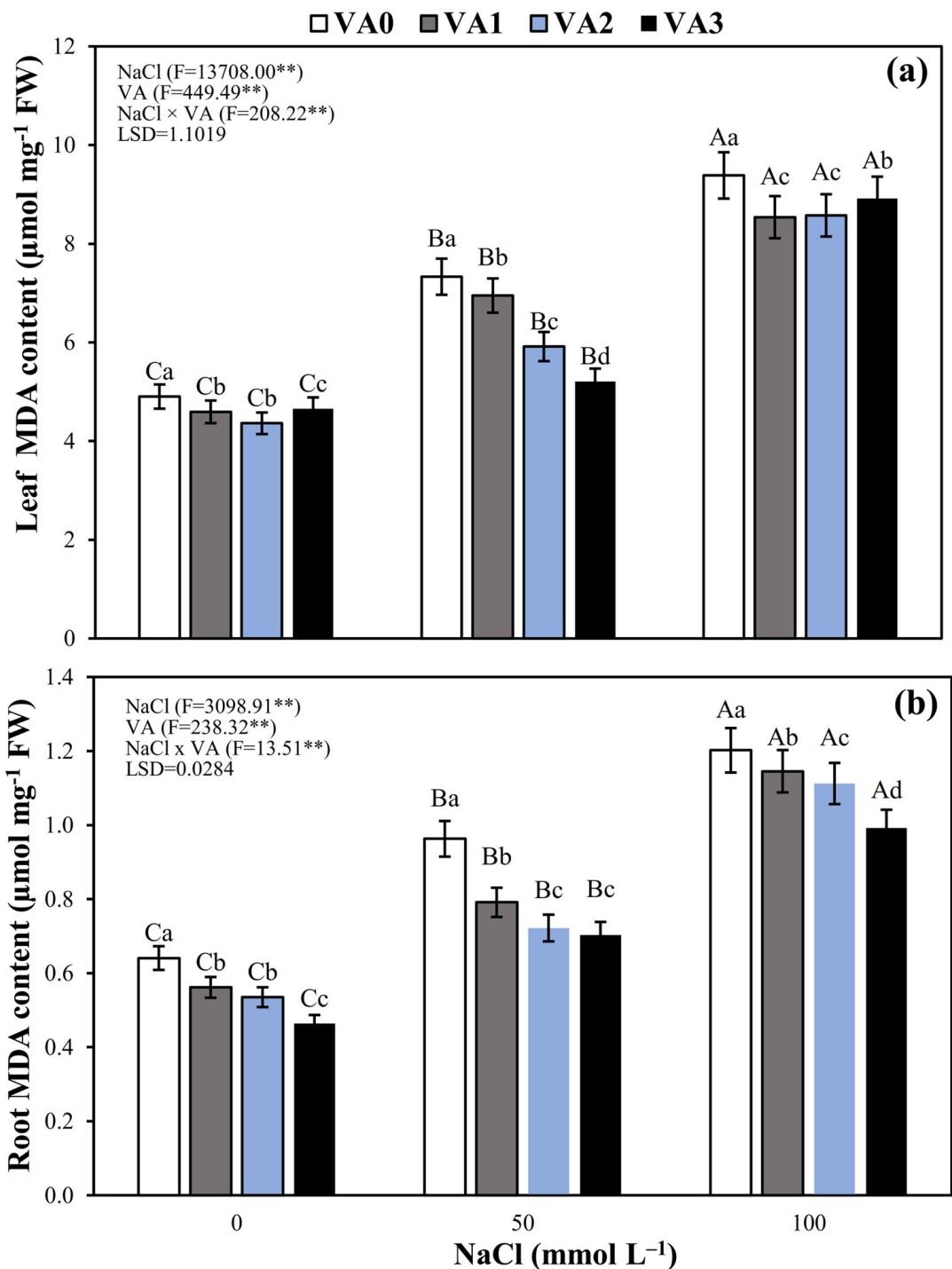

**Figure 8.** Leaf (**a**) and root (**b**) malondialdehyde (MDA) content of soybean plants under different levels of salt stress and VA application. Plants were submitted to four concentrations of VA (VA0, non-VA application; VA1, 0.4 mL L$^{-1}$; VA2, 0.8 mL L$^{-1}$; and VA3, 1.2 mL L$^{-1}$) and three levels of NaCl (0, 50, and 100 mmol L$^{-1}$). Data are mean ± SD (n = 5). Uppercase letters above bars indicate significant differences among NaCl concentrations at the same VA concentration. Lowercase letters above bars indicate differences among VA concentrations at the same NaCl concentration, according to LSD test ($p < 0.05$). F values from ANOVA: ** $p < 0.01$.

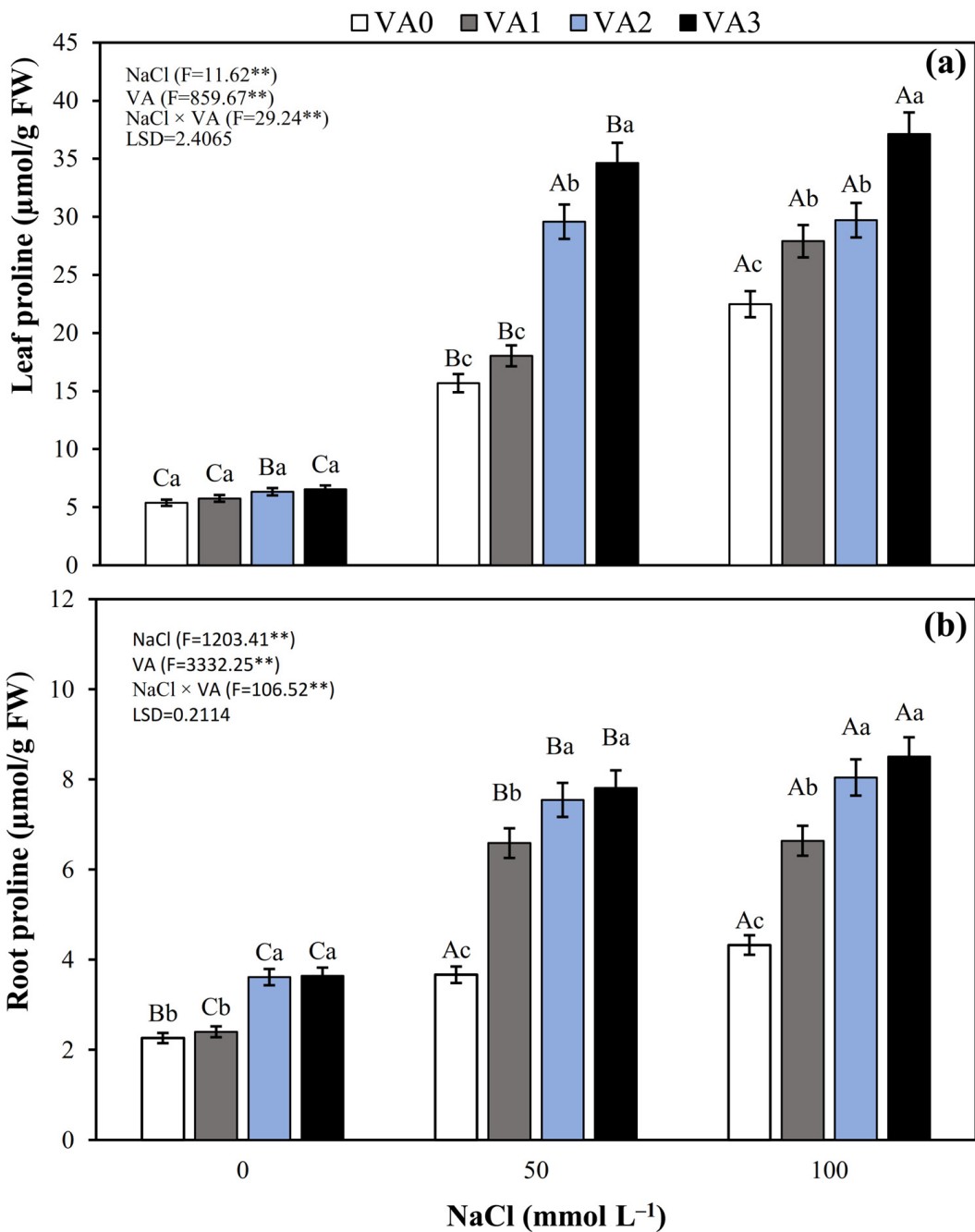

**Figure 9.** Leaf (**a**) and root (**b**) proline content of soybean plants under different levels of salt stress and VA applications. Plants were submitted to four concentrations of VA (VA0, non-VA application; VA1, 0.4 mL $L^{-1}$; VA2, 0.8 mL $L^{-1}$; and VA3, 1.2 mL $L^{-1}$) and three levels of NaCl (0, 50, and 100 mmol $L^{-1}$). Data are mean $\pm$ SD (n = 5). Uppercase letters above bars indicate significant differences among NaCl concentrations at the same amino acid concentration. Lowercase letters above bars indicate differences among amino acid concentrations at the same NaCl concentration, according to LSD test ($p < 0.05$). F values from ANOVA: ** $p < 0.01$.

## 4. Discussion

In this study, we evaluated the effects of foliar application of AA mixtures on soybean plants under different intensities of salt stress. Corroborating our two initial hypotheses, our main results showed that AA mixtures are able to mitigate the negative impacts of salt stress on soybean plants. Below, we discuss the physiological mechanisms that plants use to tolerate salt stress conditions and how AAs improve tolerance of the salt stress.

It is widely reported in the literature that soybean growth is reduced under salt stress. [15,36]. As observed here, salt stress can lead to various physiological and molecular changes, limiting plant growth by inhibiting photosynthesis and reducing available resources. Salt stress affects the formation of the light-harvesting complex and regulates the state transition of photosynthesis [37]. In our study, we observed this fact on soybean plants submitted to concentrations of 50 and 100 mmol L$^{-1}$ of NaCl (Figure 2a–d), while AA application attenuated the deleterious effects both for dry mass and for leaf area. Plants treated with AAs showed greater growth when compared with plants grown in the presence and absence of NaCl. The positive role of AAs on growth and development has been widely verified in other crops of economic importance [22,38].

Salt stress is one of the main environmental factors that impairs physiological and metabolic processes in plants, such as photosynthesis, the integrity of photosynthetic pigments, and stomatal functioning [4,39]. Therefore, plants use several mechanisms to avoid the accumulation of Na$^+$ and Cl$^-$ within their tissues. Here, we observed that when soybean plants were exposed to salinity levels of 50 and 100 mmol L$^{-1}$ of NaCl, $A$, $E$, and $gs$ were reduced. It has been reported that when plants are subjected to salinity stress, the stomata close to minimize the root uptake of Na$^+$ and Cl$^-$ [33]. Therefore, our results indicated that soybean plants reduced the $E$ flux to avoid the accumulation of NaCl. Moreover, we observed that the application of AAs attenuated the negative effects of salt stress on soybean gas exchange (Figure 3a–c).

This buffering effect of AA mixtures against the negative effects of salt stress on A is presumably related to the crucial role of AAs in protecting proteins and photosystems. For example, similar results to those observed here were obtained by [17], in which the authors showed that the photosynthetic activity of sunflower plants was improved with the application of arginine. In addition, AAs can act as important osmolytes to balance cellular osmotic potential and control ion transport and stomata opening [34]. For instance, exogenous application of tryptophan has been shown to increase $A$, $E$, and $gs$ of many different crops [35]. Therefore, we hypothesized that foliar application of AA mixtures would improve the photosynthetic tolerance of soybean plants to the effects of salt stress and revert some of the effects of salt stress on $E$ and $gs$.

Stomatal pores are not only important for plant gas exchange but also crucial for leaf cooling capacity. Plants regulate leaf temperature mainly by controlling the opening of stomatal pores. Therefore, when stomata are open it facilitates the removal of water vapor and heat from the leaf to the atmosphere, cooling the plant canopy [36]. Here, we observed that soybean plants treated with 50 and 100 mmol L$^{-1}$ of NaCl showed higher leaf temperatures compared to plants treated with no NaCl, as expected. This rise in leaf temperature can approximate leaf temperature to the optimum temperature of photosynthesis or exceed the maximum temperature of photosynthesis, leading to damage in photosynthesis [36]. Photosynthesis is a highly sensitive process at high temperatures, and PSII is considered the most sensitive component of the photosynthetic apparatus at high temperature [37]. As result, more severe damage can be amplified in soybean leaves treated under warmer and salt-stress conditions. Our data showed the treatment of plants with AAs decreased the leaf temperature under salinity conditions, presumably due to the mitigation effects on $gs$ and $E$ (Figure 3b, c). This response is corroborated by previous research, which revealed that increasing the leaf temperature of plants growing under salinity can further promote the deleterious effects of salt stress [38,39]. The results found in our study are consistent with previous research on wheat [40] and cauliflower [41], which showed that the exogenous application of AAs, such as arginine, may be a viable strategy to improve tolerance to abiotic stress.

Beyond gas exchange parameters, we analyzed how our treatments changed the structure of photosynthetic apparatus using chlorophyll fluorescence and chlorophyll index. Chlorophyll is an important pigment for the proper functioning of the photosynthetic apparatus. The change in leaf color and, mainly, the monitoring of leaf chlorophyll levels are widely used as an indicator of plant status under abiotic stress conditions [40]. Our data

showed that the leaf chlorophyll index decreased in the presence of NaCl; however, this decrease was less pronounced with the foliar application of AA mixtures (Figure 4b). When plants grow under conditions of abiotic stress, there is degradation of the chloroplast cell membrane and other organelles such as mitochondria and the endoplasmic reticulum [11].

The quantum efficiency of photosystem II is considered adequate in a range of 0.75 to 0.85, and the photochemical efficiency can be used as an indicator of the maximum efficiency of the photochemical process in photosystem II and the potential photosynthetic activity, showing, on average, normal values between 4 and 6 $\mu$mol electrons m$^{-2}$ s$^{-1}$ [41]. According to the results obtained here, both quantum and photochemical efficiency were reduced in soybean plants subjected to salt stress; however, plants that received AA application were less impacted (Figure 5a,b). This effect of AA mixtures can be explained by their potential role in scavenging ROS, thus decreasing the oxidative damage caused by salt stress in the photosynthetic apparatus. [42,43]. In addition, the effect of arginine and glycine in increasing photosynthetic pigments was also observed in corn [44], sunflower, [23] and *Pereskia aculeata* [45].

Our treatments also impacted plant–water relations. The observed RWC decrease under salt stress observed in this study indicates that plants were under osmotic stress [18,46]. As expected, RWC reduced as the NaCl concentration increased. However, this effect was attenuated by the application of AA mixtures. This effect can be explained by the increase in K$^+$ content and decrease in Na$^+$, in addition to the regulation of *A*, *E* and *gs*. On the other hand, the foliar application of AA mixtures can regulate water relations due to the regulation of ion transport [47]. These results corroborate other studies conducted in *Ocimum basilicum* [48] and tomato [38], which suggested the involvement of glycine and tryptophan in improving the iWUE of plants.

Under salt stress, plants accumulate high concentrations of Na$^+$, affecting the homeostasis of other elements such as K$^+$ and NO$^{3-}$ and leading to other physiological problems and ion imbalances [18,49]. Our results showed that despite the high accumulation of Na$^+$ in plants under 50 and 100 mmol L$^{-1}$ of NaCl, exogenous application of AAs reduced Na$^+$ content while increasing K$^+$ content in plant tissues (Figure 8a–d). Various abiotic stresses, including salinity, result in cell desiccation and ionic and osmotic imbalance. As a response against these events, plants accumulate compatible osmolytes such as sugars, proline, AAs, or proteins. [21]. Therefore, the incorporation of AAs in plant tissues may be associated with the storage of precursors for protein synthesis in order to prepare for the rapid recovery of plant metabolism after stress. [50]. Recent studies have also highlighted the importance of AAs in the regulation of cellular ionic homeostasis [51,52].

In our study, MDA content increased under salt stress conditions. Lipid peroxidation is a process that occurs in cell membranes by NaCl. Peroxidative damage results mainly from the oxidative deterioration of unsaturated fatty acids in membranes by the action of ROS, such as hydrogen peroxide, present inside the cells. [53]. This increase in lipid peroxidation is considered as the beginning of the occurrence of oxidative damage in cells. In cultures subjected to salt stress, there is a differential inhibition of the synthesis of some proteins. In addition, ROS can cause irreversible metabolic damage through the oxidation of nucleic acids, protein denaturation, and lipid peroxidation, which may result in the loss of cell viability, drastically compromising the performance and productivity of cultures. [18]. As observed here, the application of AAs decreased the MDA content as a result of lipid peroxidation caused by salt stress, confirming the role of AAs as protective molecules against oxidative damage, presumably due to the increased activity of antioxidant enzymes [20].

Plants respond to osmotic stress by accumulating large numbers of osmolytes under high salinity conditions, such as proline [4,47]. Proline is a low molecular weight cyclic AA and is one of the main osmoprotectants recognized for regulating salinity tolerance in plants, protecting membrane integrity, and stabilizing enzymes/proteins [54,55]. According to our results, AAs increased the proline content in leaves and roots in soybean plants under salt stress (Figure 7a,b), resulting in a better adaptation and osmotic adjustment

of the plants. This fact has been evidenced in other cultures in which AAs were used to mitigate the effects of salinity [56,57].

AAs are important plant metabolites in protein synthesis and other key cellular functions. Some experiments indicated that arginine and glycine play a role in physiological processes such as photosynthesis [20]. In addition, AAs can act as important osmolytes to balance cellular osmotic potential and control ion transport and stomata opening [47]. Similar results to those observed here were obtained by [23], in which the authors showed that the photosynthetic activity of sunflower plants was improved with the application of arginine. In addition, exogenous application of tryptophan has been shown to increase the $A$, $E$, and $g_s$ of many different crops [44].

In this study, we observed that soybean plants treated with 50 and 100 mmol $L^{-1}$ of NaCl showed higher leaf temperatures compared to plants treated with no NaCl. This response is expected, since we observed reduced values of $g_s$ and $E$ under salt stress treatments [58]. Plants regulate leaf temperature mainly by controlling the opening of stomatal pores. Therefore, when stomata are open, it facilitates the removal of water vapor and heat from the leaf to the atmosphere. Here, the treatment of plants with AA mixtures decreased the leaf temperature under salinity conditions, presumably due to its mitigation effects on $E$ and $g_s$ (Figure 3b,c).

Photosynthesis is a highly sensitive process at high temperatures, and PSII is considered the most sensitive component of the photosynthetic apparatus at high temperature [59]. As a result, more severe damage can be amplified in soybean leaves treated with salt stress and increased leaf temperature. This response was corroborated by previous research, which revealed that increasing the leaf temperature of plants growing under salinity can further promote the deleterious effects of salt stress [13,60]. The results found in our study are consistent with previous research on wheat [61] and cauliflower [62], which showed that the exogenous application of AAs, such as arginine, may be a viable strategy to improve tolerance to abiotic stress. Thus, this study indicated that the foliar application of AA mixtures is a viable strategy to increase the sustainability of soybean cultivation under salinity conditions.

## 5. Conclusions

Foliar application of AA mixtures is alternative technique to attenuate the adverse effects of salt stress in soybean plants. Therefore, the current study also shows that the higher AA mixture of 1.2 mL $L^{-1}$ was the most efficient concentration in promoting the tolerance of soybean plants against salt stress. Additionality, foliar spraying of AA mixtures morpho-physiologically relieved salinity-induced damages through biological processes, including increasing $K^+$ accumulation, osmolyte hyperaccumulation, photosynthetic pigment maintenance, and water status balance, thereby resulting in a considerable decrease in $Na^+$ accumulation and concentration of MDA-induced ionic and oxidative biomarkers. These findings will be valuable for further understanding the physiological and molecular mechanisms associated with the salt tolerance in plants and the potential of using AA mixtures in sustainable production systems in arid regions.

**Supplementary Materials:** The following supporting information can be downloaded at: https://www.mdpi.com/article/10.3390/agronomy12092014/s1, Table S1: Composition of Commercial amino acid solution VIUSID Agro®.

**Author Contributions:** K.P.C. and D.O.V.: investigation and original draft preparation, with input from all authors. K.P.C., D.O.V. and A.C.H. conception and design of the experiment. K.P.C. and D.O.V. performance of the experiment and collection of data. E.H. and C.A.M.: determination and evaluation of plant physiology parameters. R.D.M.P. and P.L.G. analysis of data and draft review. L.F.L.-T., G.E.A.C. and J.C.R. contribution to the materials and lab analysis. All authors have read and agreed to the published version of the manuscript.

**Funding:** This research received no external funding.



**Acknowledgments:** We acknowledge the São Paulo State University (UNESP), especially the FCAV for providing the necessary facilities for this study. The Coordination for the Improvement of Higher Education Personnel (CAPES) and the Technological Development Support Foundation of Maranhão (FAPEMA) provided graduate studentships to K.P.C (Finance Code 001) and D.O.V (Grant BPVE-00066/22), respectively. P.L.G also thanks the Brazilian National Council for Scientific and Technological Development (CNPq) for the research fellowship (Grant No. 303846/2021-6)—Brazil.

**Conflicts of Interest:** The authors declare no conflict of interest.

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
