# Peer review of "Exogenous Application of Amino Acids Mitigates the Deleterious Effects of Salt Stress on Soybean Plants"

_agronomy, doi:10.3390/agronomy12092014_

Round 1

Reviewer 1 Report

Dear Authors

 After correcting and responding to the following points, the manuscript has the potential to be accepted:

General comments:

· This manuscript is poorly written and presented. The output of this manuscript is not evident to the reader. There are various inaccuracies at the scientific and linguistic levels.

·  The statistical data analysis is not performed or presented correctly.

· Extensive English language corrections, including spelling, grammar, and punctuation, are recommended.

· The following information should be included in the writing of an experiment in the stress field:

Growth traits

Physiological characters

Biochemical traits

Specific comments:

Abstract:

L20-21 (However…..plants): This sentence should be eliminated.

L21-23 (this study, we aimed to evaluate the effects of exogenous application of amino acids on physiology, biochemistry, and growth of soybean plants submitted to salt stress.): The goal is not written correctly because when we study the effect of stress factors, we usually start with growth traits and then move on to physiological and chemical traits. The study's goal should be changed to: This study aimed to evaluate the effects of exogenous amino acid application on the growth, physiology, and biochemistry traits of salt-stressed soybean plants.

L23-24: The authors should include the name of the experimental design used in this study, as well as the levels of each factor.

L25: The leaf gas exchange results are not available in the results section.

In this section, you used the word (we) several times. Please do not repeat these words more than once.

L24-27 (During….growth): This sentence should be deleted because it conveys the same meaning as the previous sentence.

L27-31: The authors should include some lines about the percentage of study traits that are increasing or decreasing. Furthermore, the authors should include some lines about the impact of each factor individually, such as the impact of the amino acids solution on the overall experiment.

Keywords:

The keywords do not accurately represent the content of this study, so they should be changed.

Introduction:

This section contains no information about the soybean plant. The authors should include some information about soybeans and their importance.

L51: The authors should include some information about the ROS reaction under salinity stress at the end of this line.

L52-57: There are other mechanisms, such as the role of antioxidant compounds and enzymes, that should be added to the plant response mechanism.

L76: The study's goal should be changed to: This study aimed to evaluate the effects of exogenous amino acid application on the growth, physiology, and biochemistry traits of salt-stressed soybean plants.

L80-81: What does it mean that the ability to reduce salt stress will be greater at higher amino acid concentrations?

Materials and Methods:

L89: What exactly do you mean by "nutritive solution"? Please elaborate.

L93: Experimental design: Why did you use factorial randomized block design (Two-way-ANOVA-RCBD) in the greenhouse, as you mentioned in L83? Because the conditions under the greenhouse are controlled, the authors should use the factorial-CRD to analyze the data. The authors should repeat the analysis using the factorial design-CRD.

L107 (Amino acid and NaCl treatments): This section is poorly written. The authors should clearly explain this section in terms of:

When did the authors add the NaCl solution to the plant?

When was the amino acid solution added? Before, during, or after stress, and why?

The number of times an amino acid solution was applied during this experiment.

L115: Why did you apply the amino acid solution during the phenological stage? Some references are required.

L128 (measurements): Where can I find growth metrics?

The following information should be included in the order by the authors in this section:

Growth character measurement

Physiological trait measurement

Biochemical parameter measurement

L142 and L148: The authors should include the leaf positions (top, middle, and bottom) used in this experiment.

L161: The authors should separate the mineral measurement from the leaf area index, and so on. The authors should include growth traits such as leaf area, shoot dry weight, and so on at the beginning of the MM.

L171: There is a distinction between the student t-test and the LSD; please correct it and write the full name of the LSD, which stands for the pair-wise comparison of means.

Results:

Figure 2 does not depict the results of leaf gas exchange, and the leaf exchange figure is not included in this manuscript. Figures 2 and 9 are identical.

The authors only demonstrated the effect of the interaction between the two factors in all traits studied. There is no information or data available about the effect of each factor. Because the goal of this study is to determine the effect of amino acid solutions on salinity tolerance in soybean plants, the authors should add the effect of amino acid solutions and NaCl separately using a figure (for example, a box plot) or table. In other words, the authors should present their findings as follows:

Effect of amino acids solution on the growth, physiological and biochemical traits.

Effect of NaCl solutions on the growth, physiological and biochemical traits.

Effect of the combination of amino acids and NaCl solutions on the growth, physiological and biochemical traits.

Furthermore, the lettering on the bar of all figures is incorrect, and the authors should double-check them.

The authors mentioned (Lowercase letters above bars indicate differences between VA doses with each NaCl concentration according to student t-test (p0.05)) in all figure captions, but this sentence is incorrect because the T-test is not a method for means comparison. LSD is a method for comparing means.  

Why do the authors use uppercase and lowercase letters (Aa and Bb) in their LSD analysis?

Discussion:

This section is a repetition of the review literature (for example, L312-L332), with no interpretation of the results obtained in this study. As a result, the authors must interpret the study's findings.

Conclusion:

The authors should include some recommendations and future work in this field.

Best regards

Reviewer 2 Report

Introduction

Authors should add information regarding marker assistant selection (MAS) on soybean using chloride markers on soybeans.

Material and methods

Line 115-116. Add more information on foliar application, such as the first vegetative stage.

Reviewer 3 Report

The authors have presented lots of parameters which described the physiological status of plants under salt stress and positive effects of amino acids mixture administration. The results are coherent and give wide view on physiological and biochemical processes executed in response to salt stress, however, in my opinion the discussion does not provide the look on whole plant response. As the authors wrote their use three part of the plants, leaves, stems and roots, however in the discussion there is no information about root and leaf response. Moreover there is lack of some more general look on the effect on plants and the real discussion of the results with published data. There are lots of observation data on plants and their response to salt stress in literature, so the effect cause by compilation of AAs on plant response to stress could be discussed and compare to other methods applied to mitigate stress effect. Moreover it is important to look on the properties of this four amino acids used in the mixture and their role in stress tolerance. Also the introduction section gives only some basic information without explanation some of mechanism of action for amino acids – particularly glycine, arginine, aspartic acid, and tryptophan.

Some more detailed issues:

Figure 9 is duplicated as Figure 2 and 9, while figure 2 is missing.

M&M  There is no information what kind of nutritive solution were used – you just referred to Hoagland and Arnon. I advise you to give the traditional name of the solution in the main body of the text.

Figure 1 - I do not understand why MinRh is higher than maxRh, and how it is possible that maxRh at 35 DATS is higher than 100%

L115-116 - You have chosen some phenological stages, not each of them. I advise to refer to some description of the phases, as in soybean there are a few of them differing in the description. I understand that V3 is the phase when the third leave arise.

L125 - even root were used for aboveground dry mass determination?

L140 – there is a typos “potat”

L162  “e” should be and and “las” should be last

I am also a bit confused by VA abbreviation. In L95 you wrote that VA means some control without amino acids solution, while in the text you used VA in other meaning like in L182 and 185, 200 and so on. It makes difficult to understand the results description.

In L410-416 you wrote about amino acids accumulation in plants in response to stress and the positive role of them in plants tolerance, however you did not measure and did not confirm if the foliar application of amino acids in the form of VIUSID Agro® solution caused higher amino acids accumulation.

Moreover, when I read about VIUSID AGRO on their web page I found such information: “VIUSID AGRO is a solution to be dissolved in the irrigation water that contains: amino acids, potassium phosphate, vitamins and minerals (vegetable origin).” So you can not exclude the impact of other compounds of this mixture on the observed effects in plants, but you did not mention it.

In my opinion the paper is very descriptive and does not provide new information. I advise you to rewrite the discussion and try to find the biochemical and physiological link between all measured parameters.

Round 2

Reviewer 1 Report

Dear Authors

Thank you for your correction. Some errors still present in your manuscript and they should be fixed:

1. L146: This sentence is incomplete

2. L154:  The authors should add the units of measurements of all studied characters

3. In the evaluation of this manuscript in the first round, I requested the addition of some tables related to the effects of VA0, VA1, VA2, and VA3 across all concentrations of NaCl. I recommend the addition of these tables

Best regards

Reviewer 3 Report

The new version is just partially modified.

The most important issues:

-          There is still no information in the paper about composition of Commercial amino acid solution VIUSID Agro®. The authors claimed in the response that there are several different VIUSID, however found on the company web page just one VIUSID Agro, which is composed not only from amino acids. The authors must add producer specification as supplementary file. There is no possibility to discuss the role of amino acids as other molecules are also used.

-          In L201 the authors stated that MDA is expressed as nM MDA/g FW, while in Figure 8 it is μmol/mg-1. nM is a concentration, nmol is a content. What exactly did you measure? Most of the methods for MDA assessment based on reaction with TBA. In such methods you measure all the molecules which react with TBA, not only MDA, that is why it is better to present the results as TBARS (TBA reactive substances). Most of the methods measure the content of the molecules.

-          I still do not understand the use of VA. In line 123 you stated that VA means 0 ml L-1 amino acids concentration. How I should understand such term as “no-VA-treated plants” – no-(foliar spraying at 0 ml L-1)-treated plants? It still do not have sense. Please modified the abbreviation used in the manuscript and verified whole the body text.

Some minor issues:

-          L122-125 this sentence is no-informative. The number are not the concentration of amino acids, but the concertation of VIUSID Agro®. There is also no information about amino acids composition. You put them latter in Lines 140-141.

-          Conclusion – I disagree with the authors with the statement that amino acids is an excellent stress-mitigating agent. There is no information which amino acid, moreover the authors should measure final production of plats under stress and compare it to other methods to have some reason to stated as that.

Round 3

Reviewer 3 Report

The authors modyfied the new version and it could be accepted by the editors, however in my opinion the composition of the commercial VIUSID Agro should be deliver by the producer or authorized distributor. I ask the editors to take the decission as I am against publication of the paper, because of the lack of information about the composition stated by the producer. Why your information are so different from that on manufacturer page http://www.davidagroup.com/eng/viusid-agro/index.html
